elLIFE

# BAD and $K_{ATP}$ channels regulate neuron excitability and epileptiform activity

Juan Ramón Martínez-François[1]*, María Carmen Fernández-Agüera[2], Nidhi Nathwani[1], Carolina Lahmann[1], Veronica L Burnham[1], Nika N Danial[1,2]*, Gary Yellen[1]*

[1]Department of Neurobiology, Harvard Medical School, Boston, United States; [2]Department of Cancer Biology, Dana-Farber Cancer Institute, Boston, United States

**Abstract** Brain metabolism can profoundly influence neuronal excitability. Mice with genetic deletion or alteration of *Bad* (BCL-2 agonist of cell death) exhibit altered brain-cell fuel metabolism, accompanied by resistance to acutely induced epileptic seizures; this seizure protection is mediated by ATP-sensitive potassium ($K_{ATP}$) channels. Here we investigated the effect of BAD manipulation on $K_{ATP}$ channel activity and excitability in acute brain slices. We found that BAD's influence on neuronal $K_{ATP}$ channels was cell-autonomous and directly affected dentate granule neuron (DGN) excitability. To investigate the role of neuronal $K_{ATP}$ channels in the anticonvulsant effects of BAD, we imaged calcium during picrotoxin-induced epileptiform activity in entorhinal-hippocampal slices. BAD knockout reduced epileptiform activity, and this effect was lost upon knockout or pharmacological inhibition of $K_{ATP}$ channels. Targeted BAD knockout in DGNs alone was sufficient for the antiseizure effect in slices, consistent with a 'dentate gate' function that is reinforced by increased $K_{ATP}$ channel activity.
DOI: https://doi.org/10.7554/eLife.32721.001

*For correspondence:
juan_martinez-francois@hms.harvard.edu (JRM-F);
Nika_Danial@dfci.harvard.edu (NND);
gary_yellen@hms.harvard.edu (GY)

Competing interests: The authors declare that no competing interests exist.

## Introduction

Epilepsy is a very common neurological disorder in need of better therapies. Approximately one third of epilepsy patients are resistant to pharmacological treatments, and most of the current therapies can produce very severe side effects. One very effective alternative treatment is a very low carbohydrate, high fat diet—the ketogenic diet (*Hartman et al., 2007*; *Neal et al., 2009*; *Thiele, 2003*). On a ketogenic diet, fatty acids are converted to the ketone bodies β-hydroxybutyrate and acetoacetate, which circulate at millimolar concentrations in the blood. These ketone bodies provide an alternative fuel to tissues, including the brain (*DeVivo et al., 1978*; *Mergenthaler et al., 2013*; *Owen et al., 1967*; *Zielke et al., 2009*). The observation that increased ketone body metabolism produces seizure protection suggests that fuel utilization and neuronal excitability are linked, however, the mechanisms underlying this link are poorly understood (*Bough and Rho, 2007*). Some of the mechanisms proposed to underlie this metabolic seizure protection include changes in gene expression associated with decreases in glycolysis (*Garriga-Canut et al., 2006*; *Stafstrom et al., 2009*), glutamate release reduction (*Juge et al., 2010*), adenosine signaling through A1 purinergic receptors (*Masino et al., 2011*), regulation of excitability by the activity of lactate dehydrogenase (*Sada et al., 2015*), more efficient production of GABA from glutamate (*Yudkoff et al., 2008*, *2007*), stimulation of mitochondrial biogenesis (*Bough et al., 2006*), general 'positive' shifts in energy balance (*Bough et al., 2006*; *DeVivo et al., 1978*; *Pan et al., 1999*), increase of GABAergic tonic inhibition through neurosteroids (*Forte et al., 2016*), and a decrease in neuronal excitability mediated by metabolically sensitive ATP-sensitive potassium channels (*Giménez-Cassina et al., 2012*).

To study how brain metabolism can modify neuronal excitability we have used a non-dietary approach to alter brain fuel utilization. The protein BAD (BCL-2 agonist of cell death) regulates both apoptosis and glucose metabolism (*Giménez-Cassina and Danial, 2015*). When BAD's metabolic function is disrupted by genetic alteration in mice, neurons and astrocytes display decreased glucose and increased ketone body metabolism. These effects on brain metabolism produced by BAD alteration are reminiscent of those observed with a ketogenic diet. In fact, BAD-altered mice are also strongly resistant to behavioral and electrographic seizures. At the cellular level, BAD modifications that decrease glucose metabolism produce a marked increase in the activity of ATP-sensitive potassium ($K_{ATP}$) channels located in dentate granule neurons (DGNs). Thus, $K_{ATP}$ channels in the dentate gyrus, a region that is thought to limit input into the hippocampus acting as a seizure 'gate,' are well poised to mediate the effects of BAD on DGN excitability. Indeed, when $K_{ATP}$ channels are genetically ablated, seizure resistance produced by BAD modification is abolished (*Giménez-Cassina et al., 2012*).

In this study, we asked if BAD can effectively alter neuronal and network excitability by opening $K_{ATP}$ channels. In BAD knockout mice, we restored BAD expression in a fraction of the DGNs, and tested individual cells to learn whether BAD could regulate $K_{ATP}$ channel activity in a cell-autonomous manner. We also tested the consequences of altered $K_{ATP}$ channel activity for excitability of individual DGNs, which we measured using perforated patch recordings in the presence or absence of the $K_{ATP}$ channel inhibitor glibenclamide. Finally, to study the anticonvulsant effect of BAD in the entorhinal-hippocampal (EC-HC) circuit, we provoked epileptiform activity by partially blocking $GABA_A$-mediated inhibition with picrotoxin, and monitored the activity using imaging of the calcium sensor GCaMP6f. BAD ablation reduced the epileptiform activity of the EC-HC circuit, and this seizure-protective effect depended on the presence of functional $K_{ATP}$ channels. Furthermore, this in vitro seizure protection was reproduced by BAD deletion solely in DGNs, reinforcing the notion of the dentate gyrus as a seizure gate.

## Results

### The effect of BAD on $K_{ATP}$ channels is cell-autonomous

To test whether the effects of BAD on $K_{ATP}$ channels are cell-autonomous, we reconstituted BAD expression in the hippocampus of $Bad^{-/-}$ mice through intracranial injection of an adeno-associated virus (AAV). In addition to *Bad*, the AAV also contained the sequence for the red fluorescent protein mCherry to label transduced cells. To assess the effect of BAD on neuronal $K_{ATP}$ channels, we performed whole-cell patch-clamp experiments using an intracellular solution with a high ATP concentration (4 mM) (*Figure 1A*). We have previously shown that *Bad* deletion produces an increase in $K_{ATP}$ channel activity (*Giménez-Cassina et al., 2012*). This increase is manifested at the whole-cell level as an initial $K_{ATP}$ conductance that gradually decreases over the first few minutes of recording after break-in, as the intracellular solution is dialyzed with a sufficiently high ATP concentration to inhibit the initially open $K_{ATP}$ channels; we have called this phenomenon 'washdown' (*Giménez-Cassina et al., 2012*). In wild-type DGNs, because $K_{ATP}$ channels display a very low basal open probability ($NP_o$ = 0.05%), intracellular solution replacement with high ATP has no effect, so whole-cell conductance in this case remains unchanged after break-in, that is, washdown is absent in wild-type DGNs.

If BAD modification leads to an increase in $K_{ATP}$ channel activity in DGNs in a cell-autonomous manner, washdown is expected in unlabeled $Bad^{-/-}$ cells but not in mCherry-labeled BAD-reconstituted $Bad^{-/-}$ cells. Indeed, $Bad^{-/-}$ DGN whole-cell conductance displayed a gradual decrease during the first minute after break-in and then remained unchanged (*Figure 1B*). In contrast, BAD-reconstituted $Bad^{-/-}$ DGNs did not display washdown (*Figure 1—figure supplement 1*). We also performed control experiments with low ATP (0.3 mM) in the pipette solution to measure the maximal activatable $K_{ATP}$ conductance. In this case, if cells are not adversely affected by the overexpression of BAD and/or mCherry, $Bad^{-/-}$ and BAD reconstituted $Bad^{-/-}$ DGNs should display normal $K_{ATP}$ channel current 'run-up' as channels become disinhibited when ATP is dialyzed out of the cell. Indeed, both transduced and untransduced cells displayed increases of $K_{ATP}$ conductance of similar magnitudes (*Figure 1—figure supplement 2*) showing that the change in $K_{ATP}$ currents is not due to a change in channel number, but rather to a change in channel open probability. This experiment

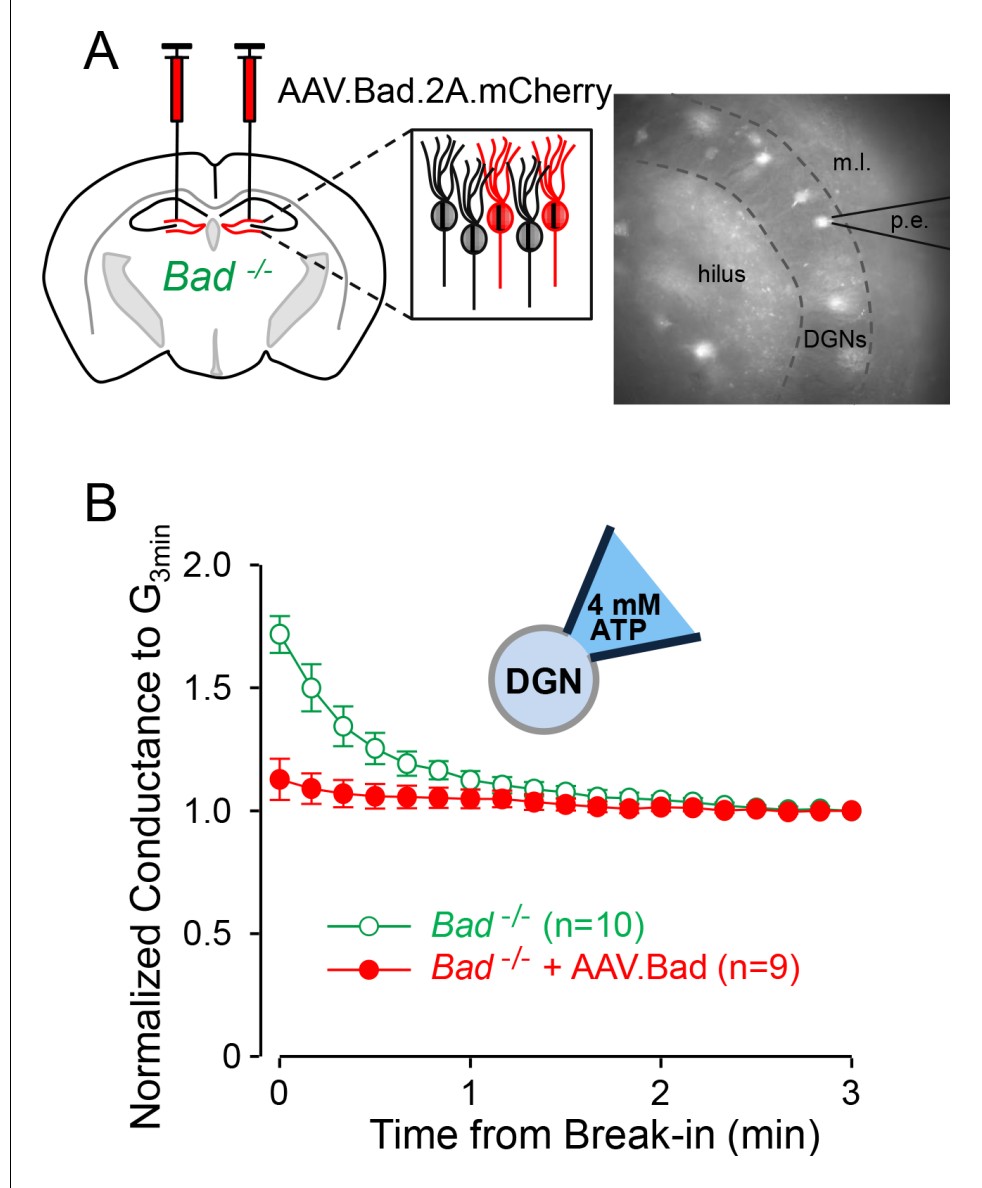

**Figure 1.** Effect of BAD on $K_{ATP}$ channels is cell-autonomous. (**A**) Left: schematic representation of intracranial injection of AAV.Bad.2A.mCherry in the hippocampus of $Bad^{-/-}$ mice to reconstitute BAD expression in mCherry labeled cells. Right: epifluorescence picture of a BAD-reconstituted $Bad^{-/-}$ DGN being recorded with a patch electrode (p.e.) in the whole-cell mode. Regions observed in picture: hilus, dentate granule cell layer (DGNs) and molecular layer (m.l.). (**B**) 'Washdown' of an initially high $K_{ATP}$ conductance, with high ATP (4 mM) in the patch electrode, seen in $Bad^{-/-}$ but not BAD-reconstituted $Bad^{-/-}$ cells. The time course of slope conductance measured during whole-cell recording was normalized to the value 3 min after break-in. Data are presented as mean ± SEM.

DOI: https://doi.org/10.7554/eLife.32721.002

The following figure supplements are available for figure 1:

**Figure supplement 1.** 'Washdown' of $K_{ATP}$ channel conductance in $Bad^{-/-}$ or BAD-reconstituted DGNs.
DOI: https://doi.org/10.7554/eLife.32721.003

**Figure supplement 2.** Total activatable whole-cell $K_{ATP}$ conductance is mostly unaffected by overexpressing BAD in $Bad^{-/-}$ DGNs.
DOI: https://doi.org/10.7554/eLife.32721.004

also confirms that $Bad^{-/-}$ and BAD reconstituted $Bad^{-/-}$ DGNs are sufficiently healthy to maintain ATP levels that keep most (if not all) of their $K_{ATP}$ channels closed. These results indicate that BAD can modulate $K_{ATP}$ channel activity in a cell-autonomous manner.

## $K_{ATP}$ channels acutely modulate dentate granule neuron excitability

We then asked whether modulation of $K_{ATP}$ channel activity could produce changes in neuronal electrical excitability. To examine DGN excitability, we performed current-clamp experiments in acute hippocampal slices. To preserve intracellular ATP dynamics, we used the perforated-patch technique to prevent dialysis of the cell's intracellular medium. In response to increasing current injections, $Bad^{-/-}$ DGNs generated fewer action potentials compared to wild-type DGNs (*Figure 2A,B*). Given that, in intact cells with physiological intracellular ATP levels, $K_{ATP}$ channels are more active in $Bad^{-/-}$ cells and almost completely inactive in wild-type cells, application of glibenclamide, a $K_{ATP}$ inhibitor, should increase excitability of $Bad^{-/-}$ neurons but produce minimal effects in wild-type. As predicted, after application of 200 nM glibenclamide, $Bad^{-/-}$ DGNs fired more action potentials, comparable to the response of wild-type DGNs (in the absence of glibenclamide). In contrast, the number of action potentials produced by wild-type DGNs was indistinguishable in the presence or absence glibenclamide (*Figure 2A,B*). These results provide evidence that $K_{ATP}$ channels can regulate DGN excitability in an acute manner. Thus, deletion of BAD could produce seizure-protective effects by decreasing neuron excitability, via modulation of $K_{ATP}$ channels.

## BAD and $K_{ATP}$ channels regulate entorhinal-hippocampal epileptiform activity

Does the decrease in neuronal excitability mediated by $K_{ATP}$ channels translate to a decrease in excitability at the network level? We have previously shown that global *Bad* deletion produces seizure resistance in vivo (*Giménez-Cassina et al., 2012*). To gain a more profound understanding of the seizure-protective mechanisms elicited by BAD knockout, we performed experiments in acute brain slices containing both the entorhinal cortex (EC) and the hippocampus. To examine the excitability of the EC-HC network, we monitored intracellular calcium dynamics in acute EC-HC slices with the fluorescent sensor GCaMP6f delivered by AAV. We elicited seizure-like events (SLEs) by application of 2.5 µM picrotoxin, a GABA$_A$ receptor antagonist. Seizure-like activity is notoriously challenging to elicit in acute slices using submersion chambers, due to lack of sufficient oxygenation (*Hájos et al., 2009*; *Hájos and Mody, 2009*; *Huchzermeyer et al., 2008*). To overcome this limitation, we used a double perfusion chamber in which EC-HC slices are exposed to the extracellular solution on both sides. The flow of the oxygen-saturated extracellular solution is sufficiently high (5 ml/min) to permit an adequate oxygenation of the slice and reliably elicit seizure-like events in the presence of picrotoxin (*Figure 3* and *Video 1*).

We observed GCaMP6f expression over the whole EC-HC area (*Figure 3A*) and detected clear seizure-like events throughout the slice, which for analysis we divided into three hippocampal areas: dentate gyrus and hilus (DGH), CA3, and CA1 and 2 EC areas: medial entorhinal cortex (MEC) and lateral entorhinal cortex (LEC). We applied picrotoxin after monitoring 10 min of baseline activity. Following picrotoxin application, SLEs started to appear in the five EC-HC regions mentioned (*Figure 3B*, top and *Video 1*). This behavior is consistent with previous reports that have used electrophysiological techniques to monitor epileptiform activity in the EC-HC network using various in vitro paradigms. We have also corroborated that the SLEs we observe by GCaMP6f fluorescence intensity changes (ΔF/F) faithfully mirror local electrical field potentials typically observed in this preparation (*Figure 3—figure supplement 1*). To analyze ΔF/F traces, we first filtered the signal to account for baseline drift, then set a threshold above which seizure-like activity was detected, and plotted the fraction of time spent in SLEs during the time of the experiment.

$Bad^{-/-}$ slices spent significantly less time in SLEs compared to wild-type slices (*Figure 3C*). Furthermore, the protection from picrotoxin-elicited epileptiform activity produced by BAD was abolished by genetic deletion of the pore-forming subunit of $K_{ATP}$ channels, Kir6.2 (*Kcnj11*), or by continuous application of the $K_{ATP}$ channel inhibitor, glibenclamide (*Figure 3C*). These observations show that *Bad* deletion prevents seizure-like activity and that $K_{ATP}$ channels acutely mediate BAD's seizure-protective effects.

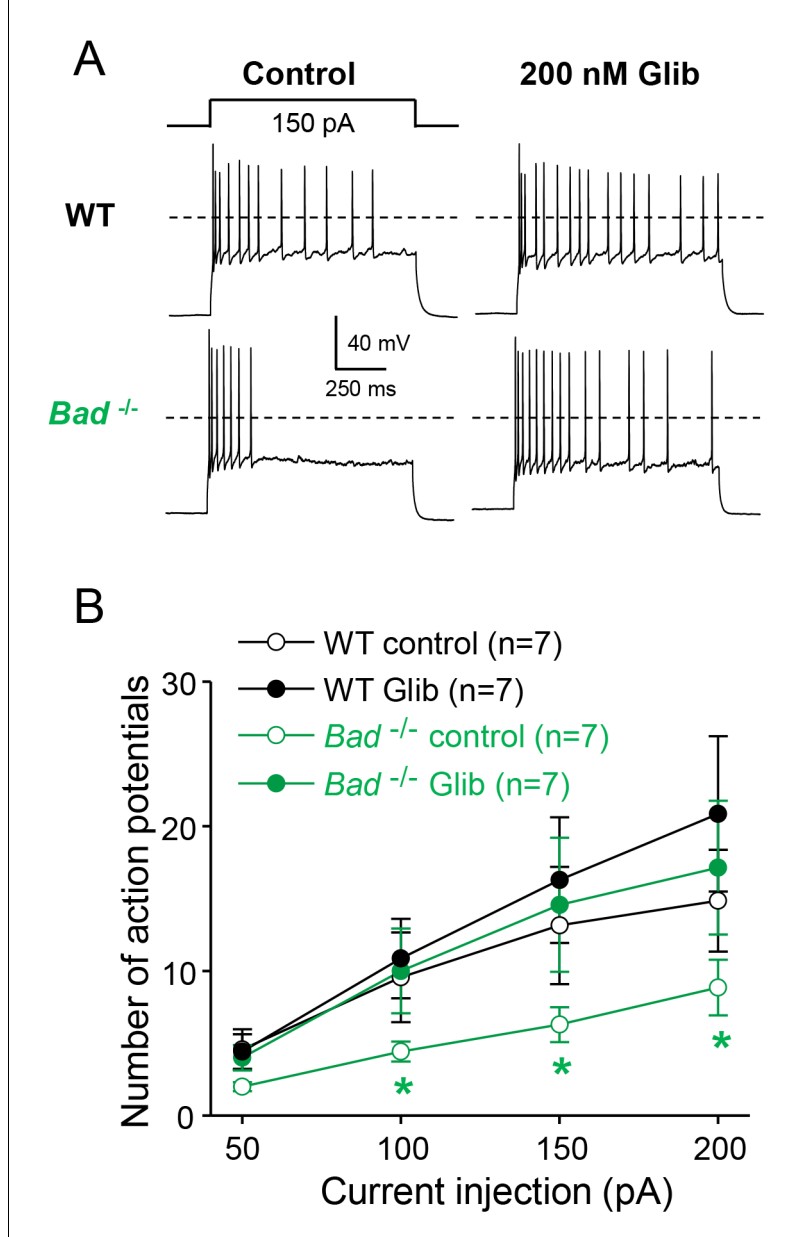

**Figure 2.** Dentate granule neurons lacking BAD are less excitable. (**A**) $Bad^{-/-}$ DGNs fire fewer action potentials that wild-type (WT), and 200 nM glibenclamide reverses $Bad$ knockout effect on DGN excitability. Representative perforated-patch voltage recordings in response to a 1 s, 150 pA current pulse for WT and $Bad^{-/-}$ DGNs, in the presence or absence of 200 nM glibenclamide, as indicated. Dotted lines indicate 0 mV. (**B**) Glibenclamide application increases firing of $Bad^{-/-}$ DGNs but not that of WT. Number of action potentials (mean ± SEM) plotted against magnitude of current injection pulse. *p<0.05; 1-way ANOVA.
DOI: https://doi.org/10.7554/eLife.32721.005

It has been shown that long-term synaptic depression mediated by postsynaptic NMDA receptors is abolished in hippocampal slices from $Bad^{-/-}$ mice (*Jiao and Li, 2011*). However, given that we observe no significant differences between the genotypes when K$_{ATP}$ channels are pharmacologically inhibited or genetically deleted, our results confirm that any genotype-specific differences in gluta-matergic transmission are not affecting the seizure phenotype we observe (see also *Giménez-Cassina et al., 2012*).

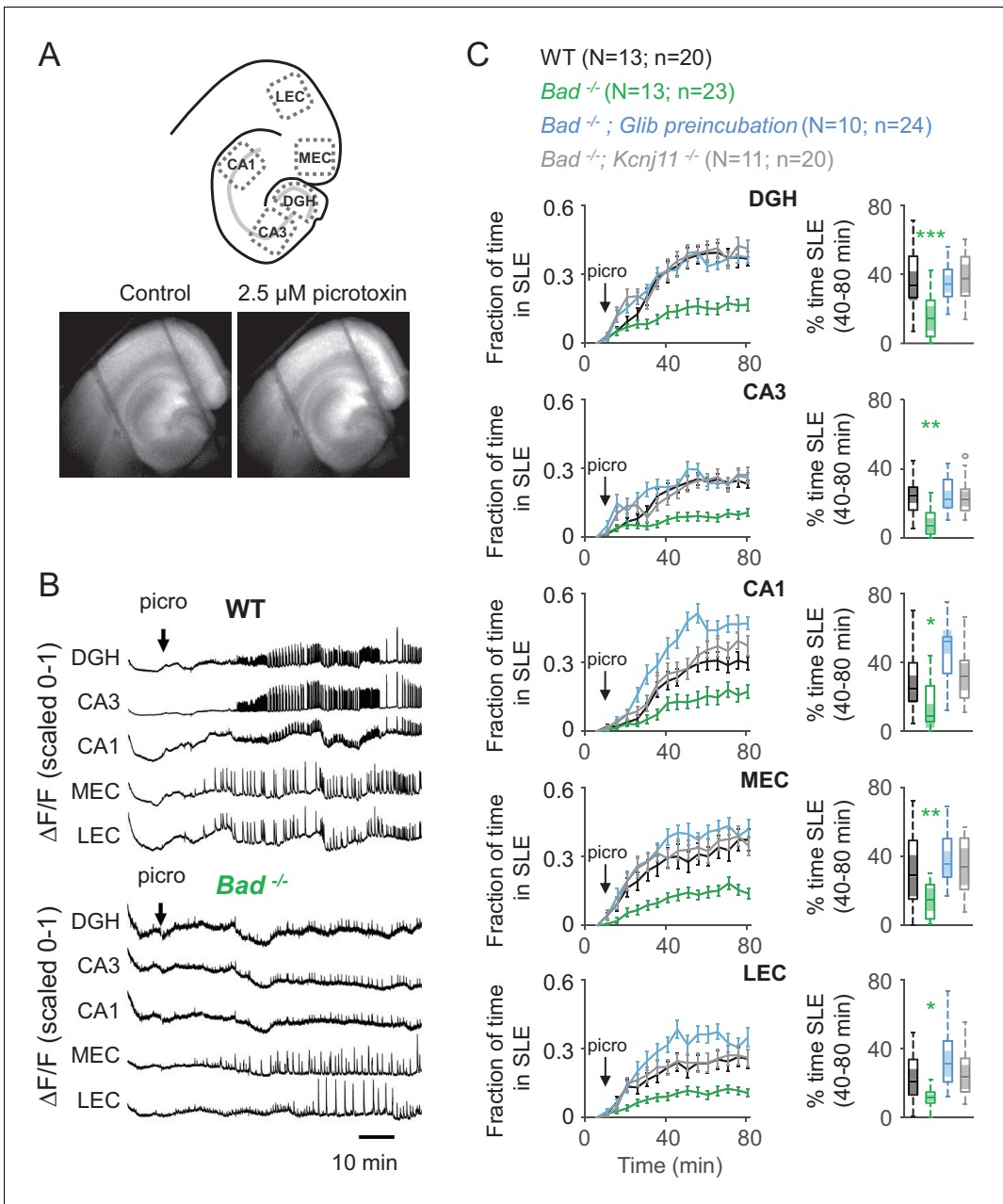

**Figure 3.** Deletion of BAD reduces picrotoxin-elicited epileptiform activity. (**A**) Top: Schematic representation of the analyzed regions of interest in EC-HC acute slices; dentate gyrus/hilus (DGH), CA3, CA1, medial entorhinal cortex (MEC) and lateral entorhinal cortex (LEC). Bottom: representative epifluorescence images of an EC-HC slice expressing GCaMP6f in ACSF (Control; left) or in the presence of 2.5 µM picrotoxin as a SLE is occurring (right). (**B**) Picrotoxin application triggers SLEs. Representative ΔF/F traces (normalized from 0 to 1) in wild-type (WT; top) or $Bad^{-/-}$ (bottom) slices for the indicated regions of interest. The arrows indicate extracellular application of 2.5 µM picrotoxin in the extracellular bath. (**C**) Genetic deletion of BAD reduced the time EC-HC slices spent in SLEs, and this effect was mediated by $K_{ATP}$ channels. Fraction of time spent in SLEs (mean ± SEM) plotted versus time (left column) or percent time in SLEs calculated from the 40–80 min range when seizure-like activity plateaus (right column) for each region of interest analyzed, and genotypes and conditions specified. *p<0.05; **p<0.005; ***p<0.0005; 1-way ANOVA.
DOI: https://doi.org/10.7554/eLife.32721.006

The following figure supplement is available for figure 3:

**Figure supplement 1.** Picrotoxin-triggered changes in GCaMP6f fluorescence intensity correspond to stereotypical epileptiform field potentials.
DOI: https://doi.org/10.7554/eLife.32721.007

Given that BAD's antiseizure effect in vivo was previously demonstrated in adult mice, we wondered if BAD-dependent protection against SLEs also occurred in adult EC-HC slices. For this, we expressed GCaMP6f in the EC-HC region through AAV delivery at age P1-2 and monitored intracellular calcium in wild-type or $Bad^{-/-}$ acute slices obtained at age P60-90 (*Figure 4—figure supplement 1*). In this case, a 20-fold higher picrotoxin concentration (50 µM) was needed to elicit a roughly similar epileptiform activity in adult EC-HC slices (*Figure 4*). This may be due to diverse factors, including the maturation of the inhibitory circuits that regulate EC-HC network excitability and maturation of DGNs themselves, which become intrinsically less excitable when fully mature in adult animals. Regardless of the picrotoxin concentration needed to elicit SLEs, deletion of *Bad* also protected against SLEs in adult EC-HC slices. There was a marked decrement in time spent in SLEs in the hippocampus, though this effect was not as strong in entorhinal areas (*Figure 4*). These results corroborate the anti-SLE effects of BAD in the adult EC-HC circuit.

## Targeted BAD knockout in the dentate gyrus reduces seizure-like activity

So far, we have observed that BAD knockout in the entire mouse leads to reduced excitability in brain slices; we proceeded to ask whether BAD knockout in a restricted subset of cells could reproduce this effect. The dentate gyrus has been proposed to act as seizure gate, preventing epileptiform activity initiated in the EC from passing onto other hippocampal areas such as CA3 or CA1. Therefore, it is possible that decreasing DGN excitability alone would be sufficient to 'strengthen' the dentate gyrus' gate function, and thus decrease excitability in the EC-HC circuit responsible for many types of seizures, such as in temporal lobe epilepsy.

To test if BAD deletion in DGNs alone is sufficient to produce a decrease in picrotoxin-induced SLEs, we produced conditional knockout BAD mice ($Bad^{flox/flox}$) and crossed them with transgenic mice (*Dock10-Cre*) that express Cre recombinase specifically in DGNs (*Kohara et al., 2014*). These mice ($Bad^{flox/flox}$; *Dock10-Cre*) have targeted knockout of *Bad* expression in the dentate gyrus but not in the rest of the hippocampus or EC, as confirmed by immunostaining (*Figure 5*, *Figure 5—figure supplement 1* and *Figure 5—figure supplement 2*). Compared to slices from both parental lines, $Bad^{flox/flox}$; *Dock10-Cre* slices (from juvenile, P13-19, mice) spent significantly less time in SLEs (*Figure 6*). Remarkably, the reduction in seizure-like activity of these slices that lack BAD solely in DGNs was comparable to that of $Bad^{-/-}$ slices (compare orange $Bad^{flox/flox}$; *Dock10-Cre* and green $Bad^{-/-}$ data in *Figure 6*). These results demonstrate that DGN-specific BAD knockout is sufficient to produce alleviation of seizure-like activity, supporting the idea that BAD's direct action in neurons can produce seizure protection. Furthermore, the results provide additional evidence to support the idea that the dentate gyrus acts as a gate for seizure activity.

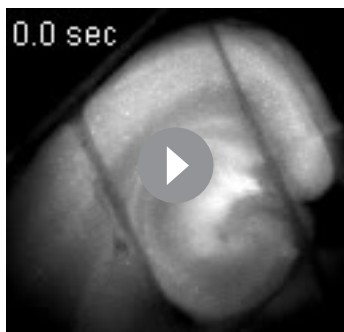

**Video 1.** Seizure-like events in the presence of 2.5 µM picrotoxin. Fluorescence intensity as imaged in *Figure 3* showing epileptiform activity in an acute EC-HP slice in the presence of 2.5 µM picrotoxin. Video was created using ImageJ 1.51 s (Fiji).
DOI: https://doi.org/10.7554/eLife.32721.008

# Discussion

## Seizure protection can be elicited by manipulation of brain metabolism

Alterations in metabolism have long been proposed as avenues to treat epileptic seizures. One example is the ketogenic diet, which has been used for almost a century as an effective treatment for epilepsy (*Hartman et al., 2007*; *Neal et al., 2009*; *Thiele, 2003*). The switch in brain fuel metabolism from using carbohydrates to ketone bodies appears to be a major contributor to seizure protection. Compared to control animals, rats on a low-carbohydrate, high-fat diet exhibit lower glucose and increased ketone body concentrations in blood. These ketotic rats show altered brain levels of metabolites, such as the whole brain ATP/ADP ratio, suggesting an

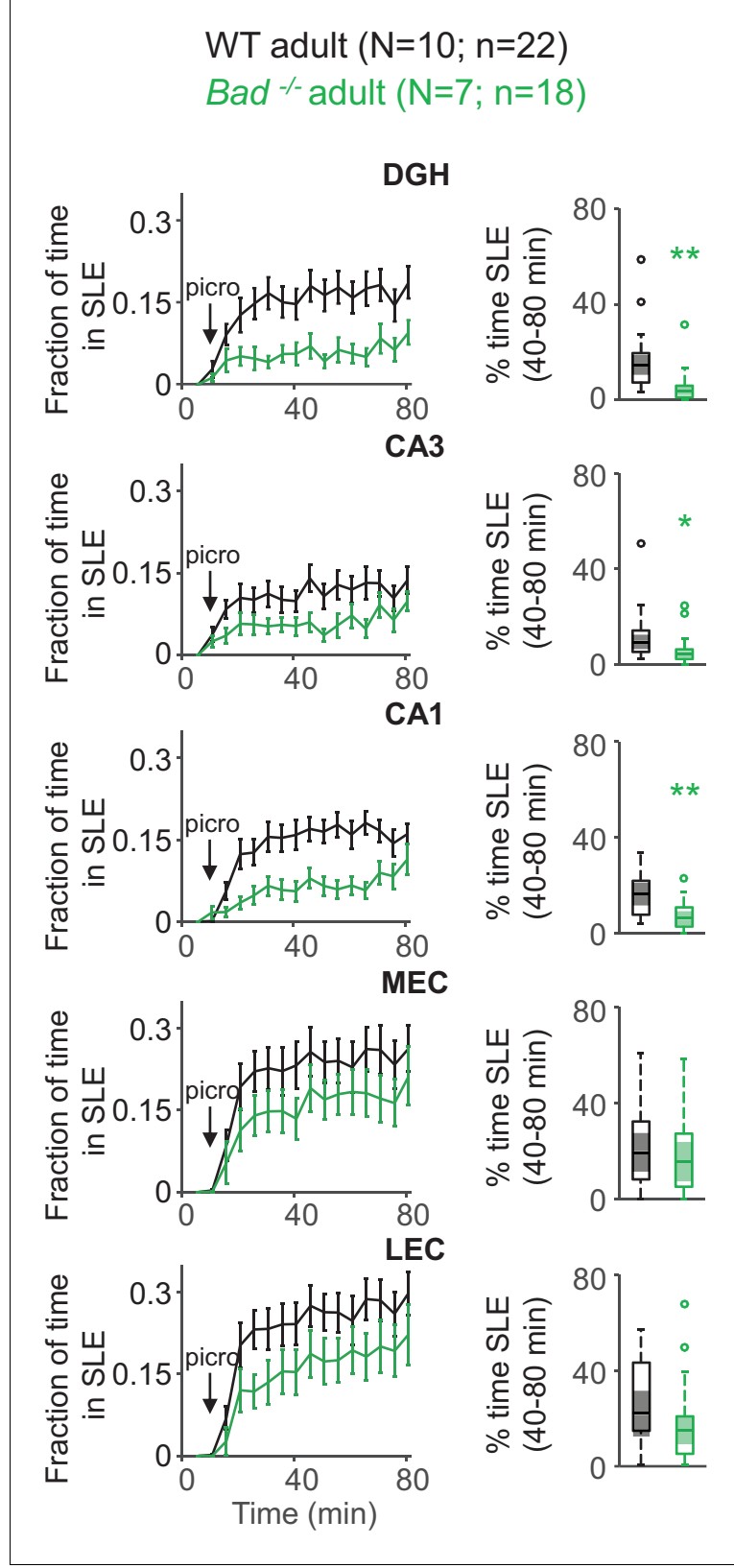

**Figure 4.** Deletion of BAD decreases picrotoxin-triggered seizure-like activity in adult EC-HC slices. Genetic deletion of BAD reduced the time hippocampal regions spent in SLEs in adult slices. Left column: fraction of time spent in SLEs (mean ± SEM) plotted versus time. The arrows indicate extracellular application of 50 μM picrotoxin
*Figure 4 continued on next page*

*Figure 4 continued*
in the extracellular bath. Right column: percent time in SLEs calculated from the 40–80 min range when seizure-like activity plateaus for each region of interest analyzed and genotypes specified. *p<0.05; **p<0.005; two-tailed Student's *t*-test.
DOI: https://doi.org/10.7554/eLife.32721.009
The following figure supplement is available for figure 4:

**Figure supplement 1.** Picrotoxin-elicited epileptiform activity in adult EC-HP slices.
DOI: https://doi.org/10.7554/eLife.32721.010

'improved' energy state (*DeVivo et al., 1978*). Further, rats on the ketogenic diet upregulate transcripts encoding mitochondrial energy metabolism enzymes (*Bough et al., 2006*). Human patients undergoing treatment with a ketogenic diet also show changes in brain energy metabolism in agreement with rodent studies (*Pan et al., 1999*). The role of this metabolic switch in producing seizure protection is supported by the fact that although the anticonvulsive effect of the ketogenic diet develops gradually over days, it can be reversed in rodents and humans very rapidly upon carbohydrate consumption (*Appleton and DeVivo, 1974*; *Huttenlocher, 1976*; *Pfeifer et al., 2008*; *Uhlemann and Neims, 1972*). A different manipulation of brain metabolism by pharmacological inhibition of glycolysis also leads to seizure protection. The glycolytic inhibitor 2-deoxy-D-glucose (2DG) suppresses rodent seizures in vivo and in vitro through changes in gene expression related to epilepsy (*Garriga-Canut et al., 2006*; *Stafstrom et al., 2009*), thus reinforcing the idea that metabolic changes can produce seizure-protective effects.

## Changes in metabolism mediated by BAD produce seizure resistance

We have used a $Bad^{-/-}$ mouse model with effects on brain metabolism that are very reminiscent of those of the ketogenic diet; this non-dietary manipulation permits the investigation of the mechanisms involved in metabolic seizure resistance. $Bad^{-/-}$ mice are resistant to acute seizures induced by either intraperitoneal injection of kainic acid, an agonist of a subclass of ionotropic glutamate receptors, or subcutaneous injection of pentylenetetrazole, an antagonist of $GABA_A$ receptors (*Giménez-Cassina et al., 2012*). Because these two pharmacologic models of *status epilepticus* act through different mechanisms, producing seizures with different characteristics, indicating that the seizure resistance produced by *Bad* deletion can be effective in different types of epilepsy. We have also shown that *Bad* deletion can reduce seizures and increase longevity in a genetic mouse model of epilepsy and sudden unexplained death in epilepsy (*Foley et al., 2018*).

BAD is a proapoptotic BCL-2 family member, but in addition to its well-known proapoptotic action, BAD also regulates glucose metabolism in various cell types (*Giménez-Cassina and Danial, 2015*). The switch between the apoptotic and metabolic roles of BAD occurs through phosphorylation of its serine 155 (equivalent to serine 118 in human BAD) (*Danial et al., 2008*). Serine 155 phosphorylation enhances glucose metabolism while inhibiting BAD's apoptotic function. Mutation of this residue to a non-phosphorylatable alanine ($Bad^{S155A}$) phenocopies the glucose-to-ketone-body fuel switch in BAD knockout neurons and astrocytes, and also produces in vivo seizure protection (*Giménez-Cassina et al., 2012*). Because both $Bad^{-/-}$ and $Bad^{S155A}$ have the same effects on $K_{ATP}$ channel activity and seizure protection, but have opposite effects on apoptosis, the effects of BAD knockout on seizure-like activity are very likely due to BAD's effect on metabolism and not on apoptosis.

Studies in mouse models with deletions in other BCL-2 family members provide further evidence that BAD-mediated seizure protection is metabolic rather than apoptotic. Knockout of BIM or PUMA leads to neuroprotection related to seizures (*Engel et al., 2011*, *2010*; *Murphy et al., 2010*). However, in contrast to our previous observations of seizure protection by BAD deletion (*Giménez-Cassina et al., 2012*), ablation of BIM or PUMA is not seizure protective immediately after kainic acid injection (*Engel et al., 2010*; *Murphy et al., 2010*). Instead, these other BCL-2 family members, promote hippocampal neuron death in a model of chronic seizures (intra-amygdala microinjection of kainic acid), thus, $Bim^{-/-}$ or $Puma^{-/-}$ animals exhibit long-term seizure protection due to decreased cell death, as opposed to $Bad^{-/-}$ mice that are resistant to acute seizures due to a switch in fuel preference.

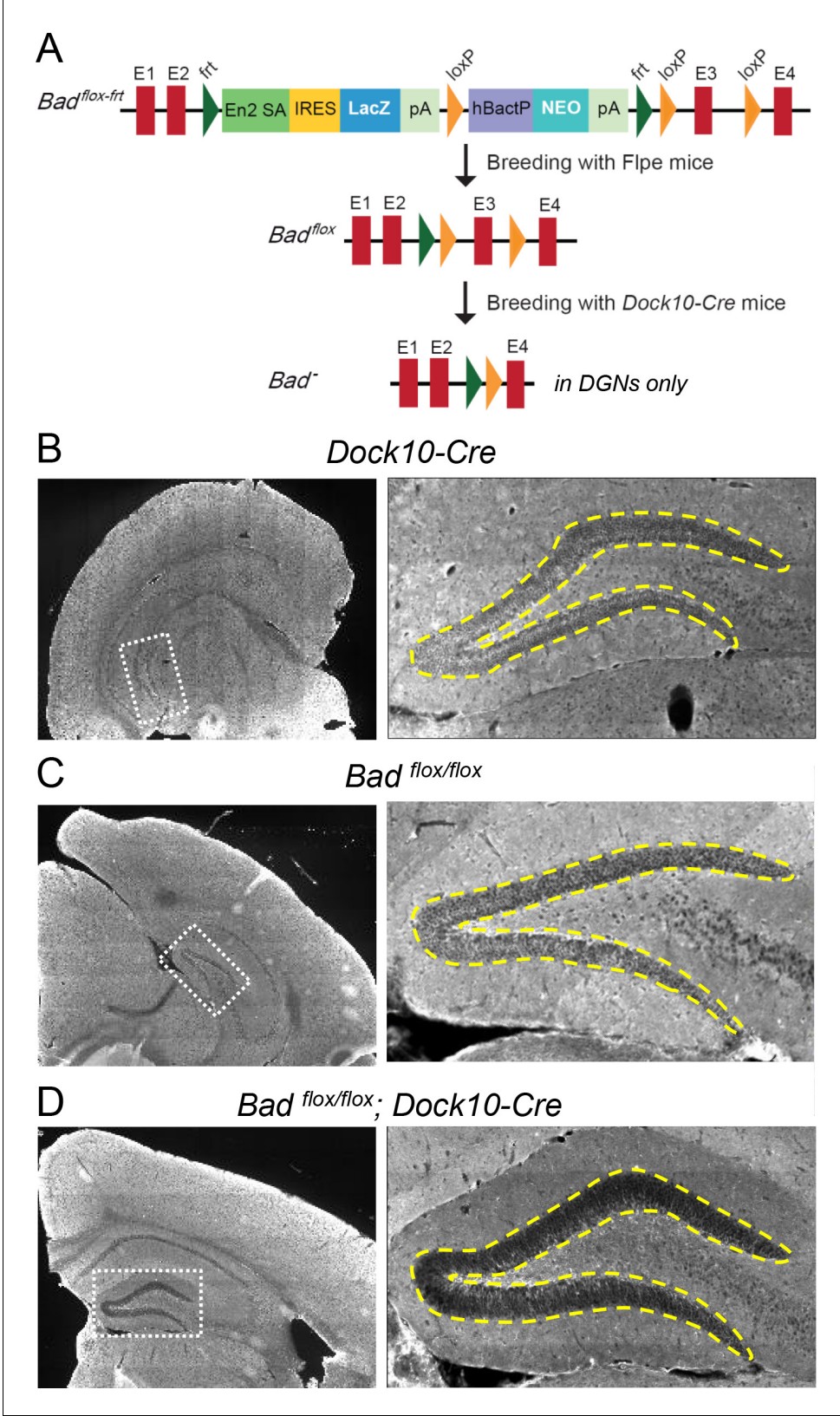

**Figure 5.** Targeted knockout of *Bad* expression in the dentate gyrus but not in the rest of the hippocampus. (**A**) Schematic illustration of the targeting construct and strategy to generate the conditional *Bad^flox^* allele. *Abbreviations*: En2 SA: En2 splice acceptor; IRES: Internal ribosomal entry site; pA: polyadenylation sequence; Frt: *Figure 5 continued on next page*

*Figure 5 continued*

Flipase (Flpe) recombinase target; loxP: Cre recombinase target; hBactP: human beta actin promoter; NEO: neomycin resistance gene. (**B, C, D**) Immunostaining for BAD in brain coronal sections from mice of the specified genotypes. Notice lack of BAD expression only in DGNs from *Bad^{flox/flox}; Dock10-Cre* brain. In the right column dashed line area demarcates the DGN cell body layer. Also notice the lower fluorescence intensity in the entire molecular layer of the dentate gyrus (containing the dendrites of the DGNs) in *Bad^{flox/flox}; Dock10-Cre* compared to the parental genotypes.

DOI: https://doi.org/10.7554/eLife.32721.011

The following figure supplements are available for figure 5:

**Figure supplement 1.** Targeted knockout of *Bad* expression in the dentate gyrus: coronal sections.
DOI: https://doi.org/10.7554/eLife.32721.012

**Figure supplement 2.** Targeted knockout of *Bad* expression in the dentate gyrus: horizontal sections.
DOI: https://doi.org/10.7554/eLife.32721.013

## Effects of BAD deletion are acutely mediated by $K_{ATP}$ channels

The metabolic shift produced by BAD knockout leads to an increase in $K_{ATP}$ channel activity. In cell-attached recordings from DGNs, where the intracellular medium—and thus ATP—is undisturbed, the open probability of $K_{ATP}$ channels is markedly increased in $Bad^{-/-}$ cells compared to wild-type cells, where these channels are mostly closed (*Giménez-Cassina et al., 2012*). Furthermore, the extra whole-cell conductance due to this increased $K_{ATP}$ channel activity in $Bad^{-/-}$ DGNs can be seen to disappear as the intracellular medium is dialyzed with an elevated ATP concentration (*Figure 1*), an effect absent in wild-type DGNs (*Giménez-Cassina et al., 2012*).

The altered $K_{ATP}$ conductance is due to the change in open probability and not to a change in the number of functional $K_{ATP}$ channels. In whole-cell experiments with low ATP in the intracellular medium, the maximal conductance measured after disinhibition of the channels is very similar in wild-type and $Bad^{-/-}$ DGNs (*Giménez-Cassina et al., 2012*) and *Figure 1—figure supplement 2*).

Furthermore, $K_{ATP}$ channels are required for the anti-seizure effect of *Bad* deletion. There is a reversal of seizure protection in $Bad^{-/-}$ mice upon further deletion of *Kcnj11* (Kir6.2), the pore-forming subunit of neuronal $K_{ATP}$ channels (*Giménez-Cassina et al., 2012*). More specifically, the susceptibility of $Bad^{-/-}; Kcnj11^{-/-}$ double knockout mice to kainic acid induced seizures is comparable to that of wild-type mice, providing compelling evidence that $K_{ATP}$ channels are required for BAD-mediated seizure resistance.

Here we found that the increased $K_{ATP}$ conductance leads to decreased DGN excitability (*Figure 2*). Previously, $K_{ATP}$ channels have been seen to affect neuronal excitability (*Allen and Brown, 2004*; *Haller et al., 2001*; *Pierrefiche et al., 1996*). $K_{ATP}$ channels link metabolism to excitability because they are inhibited by intracellular ATP, and activated by intracellular ADP (*Aguilar-Bryan et al., 1998*; *Nichols, 2006*; *Proks and Ashcroft, 2009*). They are also widely expressed in the brain (*Dunn-Meynell et al., 1998*; *Liss and Roeper, 2001*) and are present at a particularly high density in DGNs (*Karschin et al., 1997*; *Mourre et al., 1991*; *Zawar et al., 1999*), where their single-channel activity can be readily recorded (*Giménez-Cassina et al., 2012*; *Pelletier et al., 2000*; *Tanner et al., 2011*).

The regulation of neuronal excitability by $K_{ATP}$ channels has been extensively studied in the context of protection from hypoxia or ischemia: these channels prevent prolonged depolarizations that produce neuronal damage (*Sun and Feng, 2013*; *Sun et al., 2006*; *Yamada et al., 2001*). Additionally, they are crucial in the response of hypothalamic neurons to circulating glucose levels (*Ashford et al., 1990*; *Levin, 2002*; *Miki et al., 2001*; *Rowe et al., 1996*; *Song et al., 2001*). Our results show that $K_{ATP}$ channels can acutely modulate DGN excitability at the cellular level (*Figure 2*) as well as network excitability in slices that are disinhibited by picrotoxin to exhibit seizure-like events (*Figure 3*). This acute pharmacological reversal of DGN excitability supports the role of neuronal $K_{ATP}$ channels as key molecules mediating seizure protection (*Giménez-Cassina et al., 2012*; *Yamada et al., 2001*).

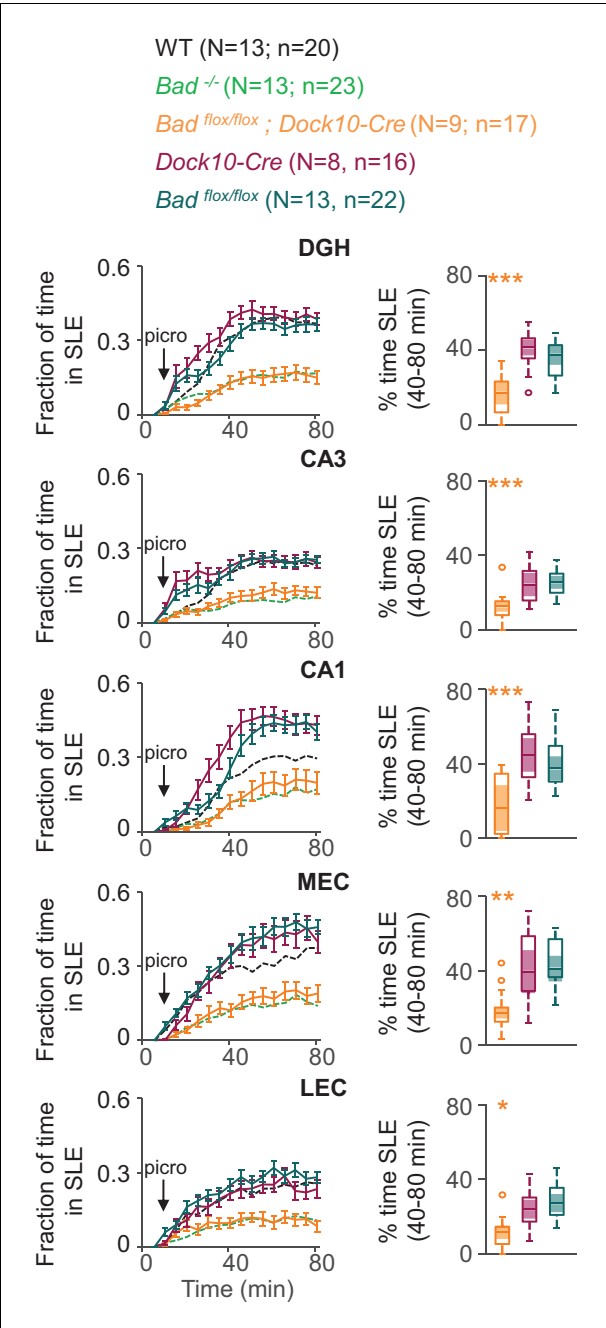

**Figure 6.** DGN-specific BAD ablation is sufficient to reduce seizure-like activity. Genetic deletion of *Bad* in DGNs alone (*Bad^{flox/flox}*; *Dock10-Cre*, orange) reduced the time hippocampal regions spent in SLEs in adult slices. Left column: fraction of time spent in SLEs (mean ± SEM) for *Bad^{flox/flox}*; *Dock10-Cre* (orange), *Dock10-Cre* (dark magenta) or *Bad^{flox/flox}* (teal) plotted versus time. For comparison, WT (black) and *Bad^{−/−}* (green) data are reproduced from *Figure 3* as dashed lines without error bars. Right column: percent time in SLEs calculated from the 40–80 min range when seizure-like activity plateaus for each region of interest analyzed. *p<0.05; **p<0.005; ***p<0.0005; 1-way ANOVA.

DOI: https://doi.org/10.7554/eLife.32721.014

## Targeted *Bad* deletion in the DGNs is sufficient to produce seizure protection

We also show that eliminating *Bad* specifically in the brain can lead to changes in $K_{ATP}$ channel activity and network excitability. When BAD was restored in single $Bad^{-/-}$ DGNs via AAV, $K_{ATP}$ channel activity reverted to that of wild-type DGNs (*Figure 1*), signifying that BAD's effect on neurons can be reversed in a cell-autonomous manner and does not seem to depend on other types of brain cells, such as astrocytes. At the network level, we show that either global knockout or DGN-targeted knockout of BAD produced a decrease in picrotoxin-induced seizure-like activity (*Figures 3* and *6*), and this effect was mediated by $K_{ATP}$ channels. This suggests that BAD's action in specific neuronal populations may alleviate seizures. However, it is yet to be tested whether *Bad* deletion solely in DGNs produces seizure protection in vivo. Future in vivo studies involving targeted *Bad* deletion in different neuron populations are needed to pinpoint whether additional brain regions are necessary for the BAD-mediated antiseizure effect.

## BAD modulates a seizure gate in the dentate gyrus

The $K_{ATP}$-mediated decrease in DGN excitability elicited inhibition of network excitability, consistent with the seizure protection observed in BAD-altered mice. The dentate gyrus controls the input from the entorhinal cortex into the rest of the hippocampus; ictal activity can be initiated in the EC and propagate through the dentate gyrus to CA3 and/or CA1. Furthermore, because of the recurrent connectivity of the EC-HP circuit due to the afferents from CA1 to the EC that pass through the subiculum, excitation can then re-enter the EC (*Andersen et al., 1966*; *Barbarosie et al., 2000*; *Barbarosie and Avoli, 1997*; *Behr et al., 1998*; *Spencer and Spencer, 1994*; *Uva et al., 2005*). In this way, the effect of the dentate gate in principle may be observed by decreased excitation not only throughout the hippocampus but also extending to the EC, as we observe in this study. Thus, limiting DGN excitability could lead to seizure protection by 'strengthening' the dentate gate. It has also been observed that ictal activity could either directly propagate from the EC to CA1 (*Ang et al., 2006*) or originate in the dentate and subsequently spread to the EC (*Lu et al., 2016*; *Meyer et al., 2016*). Our observations of a BAD-dependent decrease in time spent in SLEs are consistent with this picture (*Figures 3*, *4* and *6*). Further, we found that *Bad* deletion specific to the dentate was sufficient to elicit protection from picrotoxin-triggered SLEs. Remarkably, EC-HC slices from mice lacking BAD exclusively in DGNs displayed a similar reduction in seizure-like activity to that of slices from mice in which BAD had been globally knocked-out (*Figure 6*).

Our immunostaining results show robust BAD knockout in DGNs (including stratum lucidum, which contains the mossy fiber projections from DGNs) of $Bad^{flox/flox}$; *Dock10-Cre* mice compared to the parental controls (*Figure 5*). This targeted *Bad* knockout had no discernable effect on BAD expression elsewhere in the brain slices we examined. We cannot rule out that very sparse expression of *Dock10-Cre* in the EC (Allen Institute Mouse Brain Atlas; expression of *Dock10*) could affect some cells, however, the absence of *Bad* throughout the DGN layer of $Bad^{flox/flox}$; *Dock10-Cre* mice, together with the substantial effect on the seizure-like activity, seems most consistent with a primary effect of the dentate acting as a seizure gate, as suggested previously from both in vitro and in vivo experiments (*Dengler and Coulter, 2016*; *Heinemann et al., 1992*; *Hsu, 2007*; *Krook-Magnuson et al., 2015*; *2013*; *Lothman et al., 1992*). Interestingly, seizure-like activity is decreased in all areas (including EC) of the EC-HC slices upon BAD knockout specifically in DGNs; as previously mentioned this could reflect the recurrent nature of the EC-HC circuit where the dentate gate could decrease reentrant excitation into the EC. Taken together, these observations illuminate a mechanism by which neuronal metabolism can regulate neuronal excitability to produce seizure protection.

## BAD-mediated seizure protection remains throughout brain development

Mechanisms of both seizure production and seizure protection appear to change throughout brain development (*Dengler and Coulter, 2016*; *Holmes and Ben-Ari, 2001*). Consistent with this, we observed that a 20-fold higher picrotoxin concentration is required to elicit robust SLEs in adult slices compared to juvenile (*Figure 4*). In the juvenile mouse brain, most DGNs are immature and still integrating into the circuit; they are more excitable than mature cells and have yet to fully develop their inhibitory connections (*Espósito et al., 2005*; *Mongiat and Schinder, 2011*; *Piatti et al.,*

*2013*; *Schmidt-Hieber et al., 2004*). Our results are also consistent with developmental changes in GABAergic signaling, including the shift from a depolarizing to a hyperpolarizing chloride current during brain maturation (*Ben-Ari et al., 2012*; *Hollrigel et al., 1998*; *Liu et al., 1996*; *Rheims et al., 2008*; *Yu et al., 2013*).

We also observed that in adults, BAD-mediated seizure resistance is more limited to the hippocampal region and is not as prominent in the EC (*Figure 4*). This pattern is consistent with BAD-mediated reinforcement of a dentate gate that limits invasion of elevated EC activity into the hippocampus. The fact that in juvenile slices, DGN-specific deletion of BAD also affected EC activity may indicate that, at the lower levels of disinhibition (lower [picrotoxin]), recurrent excitation from the hippocampus to EC is more relevant than in adult slices.

## BAD knockout leads to $K_{ATP}$ channel activation by several possible mechanisms

The detailed mechanism by which BAD knockout influences $K_{ATP}$ channel activity is still unknown. One possibility is that although ATP in the whole brain of $Bad^{-/-}$ mice is not reduced (*Giménez-Cassina et al., 2012*), the ATP concentration in a compartment local to the channels is reduced. Indeed, ATP consumption at the plasma membrane by the $Na^+/K^+$ pump can increase $K_{ATP}$ channel opening (*Haller et al., 2001*; *Tanner et al., 2011*). Further, glycolytic enzymes are associated with membrane proteins (*Dhar-Chowdhury et al., 2007*; *Lu et al., 2001*; *Mercer and Dunham, 1981*; *Paul et al., 1979*) and have been found to form a complex with $K_{ATP}$ channels (*Dhar-Chowdhury et al., 2005*; *Dubinsky et al., 1998*; *Hong et al., 2011*). Glycolytic inhibition by application of 2DG produces a decrease in cellular and network excitability mediated by $K_{ATP}$ channels and an increase of GABAergic tonic inhibition through neurosteroids (*Forte et al., 2016*). Even though a direct interaction between BAD and the glycolytic enzyme hexokinase has not been studied, BAD interacts with glucokinase (hexokinase IV) in pancreatic β-cells and hepatocytes (*Danial et al., 2008*, *2003*; *Giménez-Cassina et al., 2014*), and thus, could be affecting glycolytic ATP production through this interaction. Metabolites other than ATP, such as PIP2, also regulate $K_{ATP}$ channels (*Li et al., 2017*; *Martin et al., 2017*; *Shyng et al., 2000*; *Shyng and Nichols, 1998*), and *Bad* deletion might modify the properties of $K_{ATP}$ channels by altering PIP2 or through other mechanisms, such as phosphorylation (*Béguin et al., 1999*).

## Conclusion

We have shown that $K_{ATP}$ channels can acutely regulate cellular and network excitability, but whether BAD manipulation itself can affect excitability in an acute manner remains an open question. Inhibitory molecules of BAD's glycolytic role would be valuable to learn the time course by which BAD manipulation exerts its effect on $K_{ATP}$ channels, and thus, seizure protection. Because $K_{ATP}$ channels regulate a wide variety of processes in the brain, targeting the (still unknown) mechanisms that link the BAD-induced change in metabolism to excitability might prove a better pharmacological strategy than targeting neuronal $K_{ATP}$ channels. Mimicking the effect of BAD knockout on fuel preference using small molecules could be a valuable avenue for development of new epilepsy therapies.

## Materials and methods

### Key resources table

| Reagent type (species) or resource | Designation | Source or reference | Identifiers | Additional information |
|---|---|---|---|---|
| Gene (*Mus musculus*) | *Bad* | this paper | HEPD0750_4_D04 | for generation of $Bad^{flox/flox}$ mice |
| Strain, strain background (*Mus musculus*) | male or female C57BL/6 mice | Charles River Laboratories | | |
| Genetic reagent (*Mus musculus*) | male or female homozygous $Bad^{-/-}$ mice | doi:10.1038/nm1717 | | |
| Genetic reagent (*Mus musculus*) | male or female homozygous $Bad^{-/-}$; $Kcnj11^{-/-}$ mice | doi:10.1016/j.neuron.2012.03.032 | | |

*Continued on next page*

*Continued*

| Reagent type (species) or resource | Designation | Source or reference | Identifiers | Additional information |
|---|---|---|---|---|
| Genetic reagent (*Mus musculus*) | female heterozygous *Dock10-Cre*$^{+/-}$ mice | doi:10.1038/nn.3614 | | |
| Genetic reagent (*Mus musculus*) | male or female homozygous *Bad*$^{flox/flox}$ mice | this paper | | |
| Genetic reagent (*Mus musculus*) | male or female heterozygous *Bad*$^{flox/flox}$; *Dock10-Cre*$^{+/-}$ mice | this paper | | |
| Strain, strain background (*Adeno-associated virus*) | AAV2/8.CAG.BAD.2A.mCherry | BAD-2A-mCherry plasmid packed into AAV8 at Viral Core Boston Children's Hospital | | |
| Strain, strain background (*Adeno-associated virus*) | AAV2/9.CAG.GCaMP6f | Penn Vector Core | Cat. no. AV-9-PV3081 | |
| Antibody | anti-Bad (rabbit monoclonal) | Abcam | ab32445; RRID:AB_725614 | (1:100) |
| Antibody | anti-Rabbit IgG (goat polyclonal), CF 594 | Sigma | SAB4600107 | (1:1000) |
| Other | VECTASHIELD mounting medium with DAPI | Vector Laboratories | H-1200; RRID:AB_2336790 | |
| Software | MATLAB 2016b | Mathworks | RRID:SCR_001622 | |
| Software | Fiji/ImageJ 1.51 s | NIH | RRID:SCR_002285 | |
| Software | Origin 9.1 | OriginLab | | |
| Software | Patchmaster v2x43 | HEKA | | |
| Software | TILLvisION 4.0.7.2 | Thermo Fisher Scientific | | |

## Reagents

All reagents were purchased from Sigma-Aldrich (St. Louis, MO).

## Animals

Brain slice experiments were performed using brains of male and female wild-type mice (C57BL/6; Charles River Laboratories), *Bad*$^{-/-}$ mice (**Danial et al., 2008**), *Bad*$^{-/-}$; *Kcnj11*$^{-/-}$ mice (**Giménez-Cassina et al., 2012**), *Dock10-Cre* mice (**Kohara et al., 2014**), *Bad*$^{flox/flox}$ mice (see below) or *Bad*$^{flox/flox}$; *Dock10-Cre* mice. Animals were housed in a barrier facility in individually ventilated cages with ad libitum access to standard chow diet (PicoLab 5058). All experiments were performed in compliance with the NIH Guide for the Care and Use of Laboratory Animals and the Animal Welfare Act. The Harvard Medical Area Standing Committee on Animals approved all procedures involving animals.

## *Generation of* Bad$^{flox/flox}$ *mice*

The ES cell clone bearing the targeting vector of Bad gene (HEPD0750_4_D04) was purchased from the European Conditional Mouse Mutagenesis Program, Helmholtz Center Munich, Germany (EUCOMM ID:92156). The targeted Bad allele contained an IRES:LacZ trapping cassette located 5' from a promoter-driven neomycin resistance cassette and inserted upstream of exon 3 flanked by loxP sites (knockout-first-allele design (**Skarnes et al., 2011**); see also **Figure 5A**). This allele also contained frt sites to allow simultaneous removal of LacZ and Neo cassettes by Flipase (Flpe)-driven recombination to generate the conditional *Bad*$^{flox}$ allele.

Conditional *Bad*$^{flox/flox}$ mice were generated at the Center for Genome Modification at the University of Connecticut Health, Farmington, CT. Briefly, ES cells (MJ8) were microinjected into blastocysts (BALB/c) to generate chimeric mice that were then crossed to C57BL/6N to generate *BAD*$^{flox-frt/+}$ mice. *BAD*$^{flox-frt/+}$ mice were then crossed to Flpe mice to remove the lacZ reporter and Neo resistance cassettes. *Bad*$^{flox/flox}$ mice were then mated with *Dock10-Cre* mice (**Kohara et al., 2014**) to generate *Bad*$^{flox/flox}$;*Dock10-Cre* mice in which *Bad* is selectively deleted in DGNs.

## Viral vectors

Custom-made adeno-associated virus (AAV) vector AAV2/8.CAG.BAD.2A.mCherry was obtained from the Viral Core Facility in Boston Children's Hospital, Boston, MA. AAV2/9.CAG.GCaMP6f (Cat. no. AV-9-PV3081) was obtained from the or the Penn Vector Core, University of Pennsylvania, PA.

## GCaMP6f expression in entorhinal-hippocampal area

Mice at postnatal day 1 or 2 were anesthetized by cryoanesthesia. After confirmation of anesthesia, pups were injected intracranially in both hemispheres with 200 nl of either AAV2/8.CAG.BAD.2A. mCherry or AAV2/9.CAG.GCaMP6f. The injection coordinates with respect to lambda were: 0 μm in the anterior-posterior direction, ±2.0 μm in the medial-lateral axis and -2.0 μm in the dorsal-ventral direction.

## Hippocampal slice preparation for patch-clamp recordings

Juvenile (13 to 19 days old) mice were used for all patch clamp recordings. For BAD reconstitution experiments, mice were injected (at postnatal day 1 or 2) in both hemispheres with 200 nl of AAV2/8.CAG.BAD.2A.mCherry. Mice were anesthetized by isoflurane inhalation and decapitated into ice-cold slicing solution (in mM): 87 NaCl, 2.5 KCl, 1.25 $NaH_2PO_4$, 25 $NaHCO_3$, 7 $MgCl_2$, 0.5 $CaCl_2$, 25 D-glucose, 75 sucrose and (osmolality ~340 mmol/kg). The hippocampus was isolated and mounted on a 5% agar cube which was then glued on the stage of a vibrating microtome (Campden Instuments 7000smz, Loughborough, England). Acute hippocampal slices 400 μm thick were cut in the transverse plane. Slices were then incubated at 37°C for 35 min in artificial cerebrospinal fluid (ACSF) containing (in mM): 120 NaCl, 2.5 KCl, 1 $NaH_2PO_4$, 26 $NaHCO_3$, 1.3 $MgCl_2$, 2.5 $CaCl_2$, 10 D-glucose (~300 mmol/Kg, pH 7.4). Additionally, all patch-clamp recording solutions contained 100 μM picrotoxin and 1 mM kynurenic acid to block fast synaptic transmission. These synaptic blockers were directly dissolved into ACSF immediately before experiments. After recovery, slices were stored at room temperature for 0 to 3 hr before use. All solutions were bubbled continuously with 95% $O_2$ and 5% $CO_2$.

## Entorhinal-hippocampal slice preparation for epileptiform activity recordings

Juvenile (13 to 19 days old) or adult (60 to 90 days old) mice were injected (at postnatal day 1 or 2) in both hemispheres with 200 nl of AAV2/9.CAG.GCaMP6f. Mice were anesthetized by isoflurane inhalation and then decapitated. The brain was then placed in ice-cold slicing solution (in mM): 87 NaCl, 2.5 KCl, 1.25 $NaH_2PO_4$, 25 $NaHCO_3$, 7 $MgCl_2$, 0.5 $CaCl_2$, 25 D-glucose, 75 sucrose and (osmolality ~340 mmol/kg) for juvenile mice or 93 N-Methyl-D-Glucamine, 2.5 KCl, 1.2 $NaH_2PO_4$, 30 $NaHCO_3$, 10 $MgCl_2$, 0.5 $CaCl_2$, 25 D-glucose, 20 HEPES, 5 sodium ascorbate, 2 thiourea, 3 sodium pyruvate (~320 mmol/Kg; pH 7.4 adjusted with HCl) for adult mice (*Ting et al., 2014*). Acute slices 450 μm thick were cut in the transverse plane using a vibrating microtome (Campden Instuments 7000smz, Loughborough, England). Slices were allowed to recover at 37°C for 35 min in artificial cerebrospinal fluid (ACSF) containing (in mM): 120 NaCl, 2.5 KCl, 1 $NaH_2PO_4$, 26 $NaHCO_3$, 1 $MgCl_2$, 2 $CaCl_2$, 10 D-glucose (~300 mmol/Kg, pH 7.4). After recovery, slices were stored at room temperature for 0 to 3 hr before use. All solutions were bubbled continuously with 95% $O_2$ and 5% $CO_2$. Picrotoxin was prepared daily before recordings from 50 or 500 mM stock solutions in DMSO and diluted to 2.5 or 50 μM, respectively, in recording solution. The concentration of DMSO as a vehicle control in ACSF was adjusted accordingly to 0.005% or 0.01%.

## Patch-clamp recordings

Whole-cell and perforated-patch recordings from individual dentate granule neurons (DGNs) identified under infrared differential interference contrast (IR-DIC) visual guidance were made in voltage-clamp mode following establishment of high-resistance (multi-GΩ) seals. For BAD reconstitution experiments, mCherry was excited at 580 nm using a Xenon lamp and monochromator (Polychrome IV, Thermo Fisher Scientific). Images were collected with a CCD camera (IMAGO-QE, Thermo Fisher Scientific) and TILLvisION software (Thermo Fisher Scientific). Electrophysiological data were collected with a HEKA EPC 10 patch-clamp amplifier (HEKA, Holliston, MA). Currents were filtered at 1 kHz and sampled at 10 kHz. Patchmaster v2x43 software (HEKA, Holliston, MA) was used for

amplifier control and data acquisition. Patch pipettes were pulled from borosilicate glass (VWR Scientific) using a Sutter Instruments P-97 puller. Pipette tips were fire polished to resistances of 2–3 MΩ when filled with recording solutions. During recording, ACSF was delivered to the recording bath using a peristaltic pump at a flow rate of 1 ml/min at room temperature.

For whole-cell recordings the pipette solution contained (in mM): 140 KMeS, 10 NaCl, 10 HEPES, 1 MgCl$_2$, and 0.1 EGTA (osmolality ~295 mmol/kg; pH 7.4), supplemented with 0.3 mM NaGTP and either 0.3 or 4 mM MgATP. A voltage ramp protocol was used to measure input resistance and slope conductance: from the –70 mV holding potential followed two 100 ms steps first to –100 mV then to –120 mV before initiation of a 1 s ramp from –120 to –60 mV. The prepulses were used to calculate the cell's input resistance. Voltage ramps were applied every 10 s and fitted with a straight line from –120 to –90 mV to provide a running assessment of whole-cell slope conductance.

For perforated-patch recordings, the pipette tip was filled with whole-cell pipette solution and backfilled with the same solution with amphotericin B (200 μg/ml) and Alexa Fluor 488 (10 μM; Invitrogen) added. This solution was vortexed immediately before each neuron was patched. Giga-ohm seals were achieved before a amphotericin B arrived at the pipette tip, as assessed by monitoring Alexa Fluor 488 fluorescence. The access resistance was then continuously monitored and recordings started when the access resistance was <100 MΩ. Experiments where access resistance changed >20% during the experiment or seals that took longer than 15 min to perforate were discarded. The integrity of perforated-patch recordings was judged by monitoring Alexa Fluor 488 fluorescence, and recordings where dye entered the cell were discarded. To limit recordings to a population of neurons in a homogeneous stage of DGN maturation, perforated-patch experiments were performed in DGNs with membrane capacitance of 30–40 pF. Amphotericin B was stored at –20°C in 20 mg/ml stock solutions in DMSO for one week maximum. Glibenclamide (200 nM) was diluted in ACSF from a 0.5 mM stock solution in DMSO. DMSO as a vehicle control was added to ACSF at 0.02%. Glibenclamide was applied for 10 min before recording its presence. The current clamp protocol consisted on injecting sufficient current (<30 pA) to maintain the membrane potential at –70 mV. A series of 1 s current pulses (10 s interval between pulses) were applied from 50 to 200 pA in 50 pA steps. Experiments were performed with experimenter blind to genotype.

## Patch-clamp data analysis

Data and statistical analysis were performed with Patchmaster v2x43 (HEKA, Holliston, MA) and Origin 9.1 (OriginLab, Northampton, MA, USA). The number of independent experiments (n) corresponds to the number of patched cells from different hippocampal slices. In all cases, each experimental condition was tested in slices from at least three different animals.

## Epileptiform activity recordings

In our efforts to study seizure-like activity we devised a non-electrophysiological in vitro paradigm to monitor SLEs using the fluorescent calcium sensor GCaMP6f. Epileptiform activity in the EC-HC region has been extensively studied using electrophysiological methods in acute slices (*Avoli and Jefferys, 2016*). A plethora of methods to elicit epileptiform activity in EC-HC slices have been utilized, including inhibition of potassium channels with 4-aminopyridine (*Brückner and Heinemann, 2000*; *Buckle and Haas, 1982*; *Galvan et al., 1982*; *Michelson and Wong, 1991*; *Rutecki et al., 1987*; *Voskuyl and Albus, 1985*; *Watts and Jefferys, 1993*), dishinbiting glutamate receptors by lowering the extracellular magnesium concentration (*Bragdon et al., 1992*; *Dreier and Heinemann, 1991*; *Khosravani et al., 2005*; *Mody et al., 1987*; *Walther et al., 1986*; *Whittington et al., 1995*; *Zhang et al., 1995*), or inhibition of GABA$_A$ receptors with picrotoxin (*Hablitz, 1984*; *Hablitz and Heinemann, 1989*; *Lee and Hablitz, 1989*; *Miles et al., 1984*). Compared to in vivo experiments, these seizure-like activity paradigms have several advantages, such as straightforward access to perfuse pharmacological agents and higher experimental throughput. We chose to study epileptiform activity by extracellular picrotoxin application due to the reliability of triggering SLEs slice to slice (SLEs were observed in all slices treated with picrotoxin), short latency of SLE induction and robustness of epileptiform bursts.

We sought to further increase throughput by avoiding placing electrodes in the preparation and increasing the spatial coverage of the area being monitored to the whole EC-HC region. The majority of the studies of epileptiform activity in brain slices have been performed using single-electrode

extracellular recordings, planar multielectrode arrays (*Hongo et al., 2015*; *Simeone et al., 2014, 2013*), or tetrode wire recordings (*Lévesque et al., 2016*). These methods, while having high temporal resolution, are limited to the number of electrodes that can be placed in the preparation, limiting the spatial coverage. To circumvent this constraint, in vitro epileptiform activity has also been investigated by optical methods including the monitoring of intrinsic optical signals (*Aitken et al., 1999*; *Borbély et al., 2014*; *D'Arcangelo et al., 2001*; *Weissinger et al., 2000*) or fluorescence of voltage-sensitive dyes (*Bikson et al., 2004*; *Carlson and Coulter, 2008*; *Chang et al., 2007*; *Hazra et al., 2012*; *Sinha and Saggau, 2001*). These optical methods allow for examination of seizure-like activity over a wide tissue area, however, their signal-to-noise ratio is poor and the dynamic range is very limited (a few percent), thus requiring significant signal averaging and decreasing even further temporal resolution. We expressed GCaMP6f, a fluorescent calcium sensor that has superior signal-to-noise ratio (typically a maximum of ~50% ΔF/F in our experiments) and with temporal dynamics adequate to discern picrotoxin-elicited SLEs (*Chen et al., 2013*). To monitor epifluorescence in the entirety of the EC-HC network we used a low magnification (4X) objective which makes this technique straightforward.

It is technically challenging to sufficiently oxygenate brain slices to preserve network dynamics in submerged conditions (*Hájos et al., 2009*; *Hájos and Mody, 2009*; *Huchzermeyer et al., 2008*). In a typical submerged chamber, the slices are in contact with the bath solution only on one side, thus, to achieve sufficient oxygenation, very high perfusion rates (>15 mL/min) are necessary to preserve network activity. To monitor epileptiform activity through GCaMP6f epifluorescence, we used a dual perfusion chamber, where acute slices are sitting in a grate in contact with oxygenated solution on both sides (*Hájos et al., 2009*; *Hájos and Mody, 2009*; *Lutas et al., 2014*). The dual perfusion chamber permits maintaining sufficiently high oxygenation for the generation of SLEs while also greatly facilitating imaging with an unsubmerged low-power objective to visualize the whole EC-HC area. In part this is due to perfusion rates (5 mL/min) that create a suitably undisturbed surface. This new in vitro setting greatly facilitates the study of epileptiform activity and can be easily modified to monitor other fluorescent sensors.

We recorded fluorescence intensity recordings in a dual-perfusion chamber where the slices are in contact with oxygenated ACSF on both sides continuously at a rate of 5 ml/min (2.5 ml/min/line) (*Hájos and Mody, 2009*). A slice anchor (Warner Instruments, Hamden, CT) was placed on each slice to prevent its movement during the experiment. Bath temperature was maintained at 34°C using one inline heater (Warner Instruments, Hamden, CT) per perfusion line. Solutions were kept in a water bath (VWR) at 37°C during the experiments to prevent out-gassing while being bubbled continuously with 95% $O_2$ and 5% $CO_2$. Slices were visualized with an Olympus BX51WI upright microscope using a 4X (NA 0.1 or 0.13) objective. GCaMP6f was excited at 485 nm using a Xenon lamp and monochromator (Polychrome IV, Thermo Fisher Scientific). Images were collected with a CCD camera (IMAGO-QE, Thermo Fisher Scientific) at a rate of 10 frames per second with TILLvisION software (Thermo Fisher Scientific).

Regions of interest were drawn with ImageJ (NIH) to analyze GCaMP6f fluorescence intensity changes (ΔF/F) in the entorhinal cortex or hippocampus. Subsequent analysis was performed using laboratory-made software in MATLAB (Mathworks, Natick, MA). Epileptiform activity ΔF/F time courses were filtered to account for baseline drift of the signal. Then, a threshold was set according to each trace signal-to-noise ratio and ΔF/F values above threshold were defined as seizure-like events (SLEs). Fraction of time spent in SLEs as a function of time was be plotted in 5 min bins.

## Immnunohistochemistry

Juvenile mice (16 to 25 days old) were anesthetized with 100 mg Ketamine/10 mg Xylazine i.p. per kg. Anesthesia was confirmed prior to intracardiac perfusion with PBS 1X followed by 4% paraformaldehyde (PFA) in PBS pH 7.4. Brains were immediately extracted, fixed overnight in 4% PFA at 4°C and, cryoprotected in 30% sucrose in PBS for 48 hr at 4°C, embedded in Tissue-TeK OCT (Sakura) and frozen on dry ice. Coronal or horizontal tissue sections (30 μm thick) were cut with cryostat.

Primary and secondary antibodies were prepared in blocking solution (PBS with 0.1% Triton, 10% FBS and 3 mg/ml BSA). Brain sections were treated with blocking solution for 3 hr at room temperature, incubated with BAD primary antibody (1:100, Abcam ab32445) overnight at 4°C and with secondary anti-rabbit (1:1000, Sigma SAB4600107) for 1 hr at room temperature. Tissue sections were

mounted with Vectashield Mounting Medium with DAPI (Vector Laboratories). Sections were imaged using an Olympus VS120 microscope and the secondary antibody's CF 594 fluorescence was excited with 594 nm light. For all representative images displayed in the figures, experiments were successfully repeated three or more times.

## Data and statistical analysis

The number of independent experiments (n) is indicated in each figure. For calcium imaging data, number of animals (N) is also reported. For electrophysiology experiments, sample sizes were based on previous studies (*Giménez-Cassina et al., 2012*); for calcium imaging experiments, sample sizes and power analysis were not determined in advance due to lack of knowledge of variability and likely effect sizes. In boxplots, central line indicates the median; bottom and top edges of the box indicate the 25th and 75th percentiles, respectively; whiskers extend to all data points not considered outliers; open symbols indicate outliers; and shaded area indicates 95% confidence intervals. Other summary data are presented as mean ± SEM, as indicated. Data were normally distributed and statistical significance was determined using two-tailed Student's t test or 1-way ANOVA with Bonferroni *post hoc* test as indicated. Statistical analyses were performed using MATLAB (Mathworks, Natick, MA) or Origin 9.1 (OriginLab, Northampton, MA, USA).

## Acknowledgements

We thank Sofia M. Ribeiro, Binsen Li and Hannah Zucker for assistance in the generation of AAV vectors, and members of the Danial and Yellen laboratories for critical reading of this manuscript and valuable discussions. We also thank Dr. Susumu Tonegawa for providing the *Dock10-Cre* transgenic mice; the Viral Core of Boston Children's Hospital and the U Penn Vector Core for packaging of AAVs; the GENIE project of HHMI Janelia Research Campus, Dr. Loren Looger and Dr. Douglas Kim for the GCaMP6f sensor. MCFA was supported by an EMBO long-term postdoctoral fellowship (473-2016). This work was supported by NIH grants R01 NS083844 (to NND and GY) and R01 NS055031 (to GY). This work was also assisted by core facilities supported by NIH grants P30 NS072030 (Harvard Medical School Neurobiology Imaging Facility) and P30 EY012196 (Harvard Medical School Neurobiology Machine Shop).

## Additional information

### Funding

| Funder | Grant reference number | Author |
|---|---|---|
| National Institutes of Health | R01 NS083844 | Nika Danial<br>Gary Yellen |
| National Institutes of Health | R01 NS055031 | Gary Yellen |
| European Molecular Biology Organization | 473-2016 | María Carmen Fernández-Agüera |
| National Institutes of Health | DP1 EB016985 | Gary Yellen |

The funders had no role in study design, data collection and interpretation, or the decision to submit the work for publication.

### Author contributions

Juan Ramón Martínez-François, Conceptualization, Investigation, Methodology, Writing—original draft, Writing—review and editing; María Carmen Fernández-Agüera, Investigation, Methodology, Writing—review and editing; Nidhi Nathwani, Veronica L Burnham, Investigation, Writing—review and editing; Carolina Lahmann, Supervision, Investigation, Writing—review and editing; Nika N Danial, Resources, Supervision, Funding acquisition, Writing—review and editing; Gary Yellen, Conceptualization, Supervision, Funding acquisition, Writing—review and editing

## Author ORCIDs

Juan Ramón Martínez-François  http://orcid.org/0000-0002-1035-2574
María Carmen Fernández-Agüera  http://orcid.org/0000-0002-0769-213X
Gary Yellen  http://orcid.org/0000-0003-4228-7866

## Ethics

Animal experimentation: All experiments were performed in compliance with the NIH Guide for the Care and Use of Laboratory Animals and the Animal Welfare Act. The Harvard Medical Area Standing Committee on Animals approved all procedures involving animals (protocol #03506, assurance A3431-01).

## Decision letter and Author response

Decision letter https://doi.org/10.7554/eLife.32721.017
Author response https://doi.org/10.7554/eLife.32721.018

## Additional files

### Supplementary files

• Transparent reporting form
DOI: https://doi.org/10.7554/eLife.32721.015

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
