## [Decision Letter]

[Editors’ note: this article was originally rejected after discussions between the reviewers, but the authors were invited to resubmit after an appeal against the decision.]

Thank you for submitting your work entitled "BAD and K_ATP_ channels regulate neuron excitability and epileptiform activity" for consideration by *eLife*. Your article has been reviewed by two peer reviewers, and the evaluation has been overseen by a Reviewing Editor and a Senior Editor.

Our decision has been reached after consultation between the reviewers. Based on these discussions and the individual reviews below, we regret to inform you that your work will not be considered further for publication in *eLife*.

This work clearly builds on the investigators' previous work on K-ATP channels and BAD regulation in the context of epilepsy.

Reviewer #1:

The current study "BAD and KATP channels regulate neuron excitability and epileptiform activity", is a logical extension prior work by the group (Gimenez-Cassina et al., 2012). The study uses a combination of genetic and viral vector strategies to investigate cell autonomous regulation of BAD and its effect on neuronal excitability and seizures in an in vitro model. EC-Hippocampal slices were treated with picrotoxin to induce disinhibition and generate seizure-like activity in vitro. Regional variations in seizure-like activity were examined by imaging GCaMP signals in distinct sub-regions of hippocampus and entorhinal cortex (EC). The data demonstrate cell-specific regulation of KATP currents by BAD and identify that selective BAD knockdown localized to the dentate reduces in vitro seizure-like activity in the EC-Hippocampal loop. This careful study is an important extension of prior work.

Essential revisions:

1) Disinhibition is used as the primary modality to induce seizure-like events. The experiments using glibenclamide and Kir6.2 knockouts confirm that KATP channels contribute to the reduction in seizure-like events in young BAD knockout mice. However, BAD knockouts have been shown to exhibit altered glutamatergic plasticity (Jiao and Li, 2011) and using disinhibition to generate seizures does not eliminate possibility that genotype-specific differences in glutamatergic circuits contribute to differences in seizure phenotype. Additionally, different concentrations of picrotoxin were needed to elicit seizure-like events young versus adult slices. Despite high levels of picrotoxin, seizures in adult mice appear considerably less robust suggesting that developmental changes in GABA reversal could complicate interpretation of the disinhibition model in young vs. adult mice. Thus, experiments using 4AP or 0Mg, which induce seizures without blocking inhibition, would help confirm that the primary finding reported in Figure 3 and Figure 4 can be generalized to other in seizure models.

2) The authors demonstrate that cell-specific knockdown of BAD in dentate granule cells reduces seizure-like activity in all regions of the EC-Hippo slice and propose that these data indicate that the dentate acts as a gate to the spread of seizures through the EC-Hippocampal loop. However, in the EC-Hippo slice prep, blocking seizure spread through any region within the trisynaptic loop should prevent re-entrant activity and reduce seizure duration, regardless of the specific circuit involved. Thus, in order to confirm a central role for the dentate as a seizure gate, it is necessary to demonstrate that deleting BAD in CA1 or CA3 does not similarly limit seizure activity. Again, a model where inhibition is not compromised would be appropriate for these studies as the inhibitory regulation is integral to the dentate gate.

Reviewer #2:

Martínez-François et al., describe the effects of reducing BAD levels, in particular in DG granule cells on the generation of seizure-like events (SLE) in combined entorhinal cortex/hippocampal slices exposed to the GABA-A receptor antagonist picrotoxin. The authors show that reduced BAD levels lead to the activation of K(ATP) channels, which in turn reduce cellular excitability and confer resistance to seizures. Although the ideas behind the study are sound, the execution of it leaves a lot to be desired.

1) The problem in general is the induction of SLE in slices. The authors use large-scale intracellular Ca^2+^ change measurements in various subfields of the slices as a proxy for recording electrical events. The ROIs are quite large and the neuronal populations showing the Ca^2+^ changes are quite heterogeneous. Considering that isolated CA3 and CA1 regions alone show SLE when exposed to picrotoxin, it is difficult to understand how the increase in K(ATP) activity secondary to regional deletion of BAD in dentate granule neurons can reduce the occurrence of SLE in the two hippocampal regions to the levels of the global BAD KOs. Moreover, it is inexplicable that the dentate gate effect manifests itself even to the upstream regions (MEC and LEC) when these regions are exposed to picrotoxin. A better approach would have been to expose only the EC to some convulsant drug without letting it propagate to the DG and CA3/1. An even better approach would have been to test the effects of knocking out BAD in the DG neurons in a chronic model of epilepsy. Only under the latter circumstances can meaningful conclusions be drawn about the role of BAD/K(ATP) in DG granule neurons on seizure generation and propagation.

2) The immunohistochemical evidence for BAD deletion solely in the DG granule neurons is very poor. It is not clear what method for visualization was used for the secondary Ab, but the panels (B-D) in Figure 5 are far from convincing for a specific KO of BAD.

3) Since quite large changes in K currents are reported, it would help to see some raw data on the ramps that yielded the measurements.

4) Glibenclamide is sometimes used at a concentration of 100 nM, other times at 200 nM. There is no mention in the methods if the DMSO concentration was adjusted in the vehicle controls.

[Editors’ note: what now follows is the decision letter after the authors submitted for further consideration.]

Thank you for resubmitting your work entitled "BAD and K_ATP_ channels regulate neuron excitability and epileptiform activity" for further consideration at *eLife*. Your revised article has been favorably evaluated by Richard Aldrich (Senior editor) and a Reviewing editor.

The manuscript has been evaluated by a new reviewing editor who identifies some remaining issues that need to be addressed before acceptance, as outlined below:

I think this is a very interesting manuscript. It does follow strongly on the previous study (Giménez-Cassina et al., 2012), but it shows new and more mechanistic attributes of many of the ideas inferred in that study and goes on to show that BAD deletion in the dentate is sufficient to reduce epileptiform activity, and that the mechanism is through increased K(ATP) currents.

I disagree with aspects of the previous reviews. The criticisms were with the mode of inducing epileptiform activity, but these concerns seem orthogonal to the main line of the investigation. As far as I can tell, the question being examined is, first, whether K(ATP) channels really are the pivot through which BAD manipulation exerts its effect. The previous published work showed an increase in K(ATP) single channels in BAD -/- cells, which was inferred to be a possible mechanism of seizure suppression in BAD KOs, but here we see the role of K(ATP) current investigated systematically, all the way to the effect of current blockade on firing in normal and BAD-lacking cells, which is crucial to understanding the mechanism regardless of the means by which seizure-like events are ultimately elicited. This is the first half of the manuscript, and the previous reviewers had no concerns with this. The second half of the manuscript tests whether K(ATP) modulation (including only in dentate gyrus) is the means by which BAD reduction is suppressive of seizure-like events. I know that people in the epilepsy field are highly focused on the epilepsy model used (and probably rightly so) but the previous work already showed that BAD KO was seizure suppressive both in kainate models and in disinhibitory models, so this manuscript is really beyond the realm of epilepsy models and is digging toward mechanism. So here, I think it is more than adequate to use a disinhibitory model and demonstrate that a BAD -/- mediated K(ATP) current increase (in the dentate gyrus alone) is a sufficient switch to suppress seizure-like events.

The only part that I agree with reviewer 2 on is that the authors could provide some further commentary on the fact that BAD suppression in the DG reduces seizure like events in upstream regions (MEC and LEC). I acknowledge their statement that 'the fact that the results are surprising does not invalidate them' but the authors can help us think about what the results mean. Is it possible that there is some additional BAD knockdown in the EC? I think that is what reviewer 2 was driving at with the concern about Figure 5. Or might the result imply some recurrent activity or feedback in the slice? Perhaps the authors can expand on the brief statement in the Discussion section, which may contain an answer, but is somewhat cryptic at present. I can see how the reviewer's comment made them dig in their heels but that does not do a great service to the science in the long term. A more thorough discussion of this point for a sincerely interested reader would be sufficient.

---

## [Author Response]

[Editors’ note: the author responses to the first round of peer review follow.]

Reviewer #1:The current study "BAD and KATP channels regulate neuron excitability and epileptiform activity", is a logical extension prior work by the group (Gimenez-Cassina et al., 2012). The study uses a combination of genetic and viral vector strategies to investigate cell autonomous regulation of BAD and its effect on neuronal excitability and seizures in an in vitro model. EC-Hippocampal slices were treated with picrotoxin to induce disinhibition and generate seizure-like activity in vitro. Regional variations in seizure-like activity were examined by imaging GCaMP signals in distinct sub-regions of hippocampus and entorhinal cortex (EC). The data demonstrate cell-specific regulation of KATP currents by BAD and identify that selective BAD knockdown localized to the dentate reduces in vitro seizure-like activity in the EC-Hippocampal loop. This careful study is an important extension of prior work.

We are glad that the reviewer agrees with our central conclusions and recognizes the importance of the current work.

Specific comments:1) Disinhibition is used as the primary modality to induce seizure-like events. The experiments using glibenclamide and Kir6.2 knockouts confirm that KATP channels contribute to the reduction in seizure-like events in young BAD knockout mice. However, BAD knockouts have been shown to exhibit altered glutamatergic plasticity (Jiao and Li, 2011) and using disinhibition to generate seizures does not eliminate possibility that genotype-specific differences in glutamatergic circuits contribute to differences in seizure phenotype.

We now mention the earlier work describing reduced LTD in the BAD knockouts (subsection “BAD and K_ATP_ channels regulate entorhinal-hippocampal epileptiform activity”). We also point out that our results actually do rule out a contribution of such an effect to the seizure phenotype in slices, since blockade or deletion of KATP channels completely abolishes the difference in phenotype (and any alteration in glutamatergic circuits should remain). Additionally, because disinhibition leaves glutamatergic signaling intact, it should be the ideal approach to reveal such effects.

Additionally, different concentrations of picrotoxin were needed to elicit seizure-like events young versus adult slices. Despite high levels of picrotoxin, seizures in adult mice appear considerably less robust suggesting that developmental changes in GABA reversal could complicate interpretation of the disinhibition model in young vs. adult mice. Thus, experiments using 4AP or 0Mg, which induce seizures without blocking inhibition, would help confirm that the primary finding reported in Figure 3 and Figure 4 can be generalized to other in seizure models.

The seizure-like activity in adult mice is quite robust with the higher levels of picrotoxin, as is the effect of BAD knockout. We now show examples of the primary data in Figure 4—figure supplement 1. We have discussed the likely reasons that the higher picrotoxin concentration is necessary (as the reviewer suggests, developmental changes in GABAergic signaling are likely the reason). We do not agree, however, that the interpretation is complicated by the use of disinhibition: in fact, our results show that the effects of BAD knockout are robust in the adult animals as well as the juveniles.

We considered the use of 4AP or 0Mg models, as described in the text, but found them to be less consistent at producing seizure-like behavior (and they also frequently produce spreading depression). While repeating all of our experiments with one of these models would indeed confirm that the results “can be generalized”, we would argue that these very extensive new experiments are unnecessary for the conclusions of the paper.

2) The authors demonstrate that cell-specific knockdown of BAD in dentate granule cells reduces seizure-like activity in all regions of the EC-Hippo slice and propose that these data indicate that the dentate acts as a gate to the spread of seizures through the EC-Hippocampal loop. However, in the EC-Hippo slice prep, blocking seizure spread through any region within the trisynaptic loop should prevent re-entrant activity and reduce seizure duration, regardless of the specific circuit involved. Thus, in order to confirm a central role for the dentate as a seizure gate, it is necessary to demonstrate that deleting BAD in CA1 or CA3 does not similarly limit seizure activity. Again, a model where inhibition is not compromised would be appropriate for these studies as the inhibitory regulation is integral to the dentate gate.

We have been carefully conservative in our conclusions about the dentate gate. We show (as the reviewer agrees) that targeted knockout of BAD in DGNs can produce the full antiseizure effect of BAD in the slice. We have said that these data are consistent with previous literature about the possible role of the dentate in gating seizure activity, because BAD knockout in DGNs is *sufficient* to produce an anti-seizure effect. We certainly do not claim an *exclusive* role of the dentate in seizure gating (which would require the extensive new experiments that the reviewer mentions).

Interestingly, this reviewer argues that blocking seizure spread anywhere in the region should block seizure-like events, while the other reviewer points out that other isolated regions of hippocampus exposed to picrotoxin can sustain seizure-like events (and therefore that it should be impossible for a dentate-specific manipulation to have the effect we observe). Clearly there is disagreement about what is “expected” in this model. But we would argue that this makes the very clear outcome of our experiments even more remarkable.

Additionally, we do not agree that a “model where inhibition is not compromised” would provide better evidence that the dentate can act as a gate. According to the reviewer, inhibitory regulation is central to the dentate gate, and thus we have probably produced seizure-like events by compromising the function of the dentate gate. It therefore appears from our results that we have successfully restored the gate function of the dentate by knocking out BAD and increasing KATP channel activity in dentate granule neurons.

Reviewer #2:Martínez-François et al., describe the effects of reducing BAD levels, in particular in DG granule cells on the generation of seizure-like events (SLE) in combined entorhinal cortex/hippocampal slices exposed to the GABA-A receptor antagonist picrotoxin. The authors show that reduced BAD levels lead to the activation of K(ATP) channels, which in turn reduce cellular excitability and confer resistance to seizures. Although the ideas behind the study are sound, the execution of it leaves a lot to be desired.

We are glad that the reviewer agrees with our central results.

1) The problem in general is the induction of SLE in slices. The authors use large-scale intracellular Ca^2+^ change measurements in various subfields of the slices as a proxy for recording electrical events. The ROIs are quite large and the neuronal populations showing the Ca^2+^ changes are quite heterogeneous.

Most previous work on SLEs in slices uses several individual extracellular electrodes to record the events. We think it is clear that the widefield Ca measurements provide similar (actually more extensive) information about patterns of activity, and we demonstrate that the Ca events coincide with extracellular recordings (Figure 3—figure supplement 1). In any case, the reviewer does not make it clear why this could be a problem for our conclusions.

Considering that isolated CA3 and CA1 regions alone show SLE when exposed to picrotoxin, it is difficult to understand how the increase in K(ATP) activity secondary to regional deletion of BAD in dentate granule neurons can reduce the occurrence of SLE in the two hippocampal regions to the levels of the global BAD KOs. Moreover, it is inexplicable that the dentate gate effect manifests itself even to the upstream regions (MEC and LEC) when these regions are exposed to picrotoxin.

In contrast with the earlier reviewer, this reviewer finds it very surprising that increased KATP channel activity in DGNs alone can produce the effect we observe on whole slice seizure-like activity. But the fact that our results are surprising (“difficult to understand” or “inexplicable”) does not invalidate them in any way. (And the other reviewer certainly did not find the outcomes to be inexplicable.) We believe that the results are quite clear, and the reviewer has not said otherwise.

A better approach would have been to expose only the EC to some convulsant drug without letting it propagate to the DG and CA3/1.

It is not clear why this (unusual) approach would be better, except perhaps to give a result that would be more in line with the reviewer’s expectations.

An even better approach would have been to test the effects of knocking out BAD in the DG neurons in a chronic model of epilepsy. Only under the latter circumstances can meaningful conclusions be drawn about the role of BAD/K(ATP) in DG granule neurons on seizure generation and propagation.

We have recently shown that BAD knockout indeed has antiseizure effects in (whole) mice with the *Kcna1*^−/−^ genetic model of epilepsy. But it would not be practical to perform the slice level experiments with this model, because the frequency of events would likely be very variable between animals and much lower than with the picrotoxin approach used here. Moreover, we disagree that this major new experimental campaign is necessary to make meaningful conclusions.

2) The immunohistochemical evidence for BAD deletion solely in the DG granule neurons is very poor. It is not clear what method for visualization was used for the secondary Ab, but the panels (B-D) in Figure 5 are far from convincing for a specific KO of BAD.

We disagree that the evidence is poor, and are not sure why the reviewer thinks that our method is unclear, as it is completely described in the paper.

Immunofluorescence of the hippocampus needs to be examined carefully, because the high density of nuclei in the DG and pyramidal cell layers causes them always to appear partially unstained. But careful examination of the images makes it very clear that the cytosolic signal present in the controls is absent in the targeted knockout, in the dentate granule neurons but not in other parts of the hippocampus. We have provided additional images in Figure 5—figure supplement 1. We also point out that the BAD signal is noticeably diminished in the molecular layer of the dentate gyrus (which contains the DGN dendrites), but not elsewhere in the slice or in the parental controls.

3) Since quite large changes in K currents are reported, it would help to see some raw data on the ramps that yielded the measurements.

These data are now shown in Figure 1—figure supplement 1 and Figure 1—figure supplement 2.

4) Glibenclamide is sometimes used at a concentration of 100 nM, other times at 200 nM. There is no mention in the methods if the DMSO concentration was adjusted in the vehicle controls.

This was a typographical error in one of the figure legends, and it has been corrected.

[Editors' note: the author responses to the re-review follow.]

[…] The only part that I agree with reviewer 2 on is that the authors could provide some further commentary on the fact that BAD suppression in the DG reduces seizure like events in upstream regions (MEC and LEC). I acknowledge their statement that 'the fact that the results are surprising does not invalidate them' but the authors can help us think about what the results mean. Is it possible that there is some additional BAD knockdown in the EC? I think that is what reviewer 2 was driving at with the concern about Figure 5. Or might the result imply some recurrent activity or feedback in the slice? Perhaps the authors can expand on the brief statement in the Discussion section, which may contain an answer, but is somewhat cryptic at present.

We have added Figure 5—figure supplement 2 and the following paragraphs to the Discussion section to expand on these issues:

“Our immunostaining results show robust BAD knockout in DGNs (including stratum lucidum, which contains the mossy fiber projections from DGNs) of Bad^flox/flox^; Dock10-Cre mice compared to the parental controls (Figure 5). This targeted Bad knockout had no discernable effect on BAD expression elsewhere in the brain slices we examined. We cannot rule out that very sparse expression of Dock10-Cre in the EC (Allen Institute Mouse Brain Atlas; expression of Dock10) could affect some cells, however, the absence of Bad throughout the DGN layer of Bad^flox/flox^; Dock10-Cre mice, together with the substantial effect on the seizure-like activity, seems most consistent with a primary effect of the dentate acting as a seizure gate, as suggested previously from both in vitro and in vivo experiments (Dengler and Coulter, 2016; Heinemann et al., 1992; Hsu, 2007; Krook-Magnuson et al., 2015, 2013; Lothman et al., 1992). Interestingly, seizure-like activity is decreased in all areas (including EC) of the EC-HC slices upon BAD knockout specifically in DGNs; as previously mentioned this could reflect the recurrent nature of the EC-HC circuit where the dentate gate could decrease reentrant excitation into the EC. Taken together, these observations illuminate a mechanism by which neuronal metabolism can regulate neuronal excitability to produce seizure protection.”

“We also observed that in adults, BAD-mediated seizure resistance is more limited to the hippocampal region and is not as prominent in the EC (Figure 4). This pattern is consistent with BAD-mediated reinforcement of a dentate gate that limits invasion of elevated EC activity into the hippocampus. The fact that in juvenile slices, DGN-specific deletion of BAD also affected EC activity may indicate that, at the lower levels of disinhibition (lower [picrotoxin]), recurrent excitation from the hippocampus to EC is more relevant than in adult slices.”